# Metformin with Temozolomide for Newly Diagnosed Glioblastoma: Results of Phase I Study and a Brief Review of Relevant Studies

**DOI:** 10.3390/cancers14174222

**Published:** 2022-08-30

**Authors:** Makoto Ohno, Chifumi Kitanaka, Yasuji Miyakita, Shota Tanaka, Yukihiko Sonoda, Kazuhiko Mishima, Eiichi Ishikawa, Masamichi Takahashi, Shunsuke Yanagisawa, Ken Ohashi, Motoo Nagane, Yoshitaka Narita

**Affiliations:** 1Department of Neurosurgery and Neuro-Oncology, National Cancer Center Hospital, Tokyo 104-0045, Japan; 2Department of Molecular Cancer Science, Faculty of Medicine, Yamagata University, Yamagata 990-9585, Japan; 3Department of Neurosurgery, Graduate School of Medicine, The University of Tokyo, Tokyo 113-8655, Japan; 4Department of Neurosurgery, Faculty of Medicine, Yamagata University, Yamagata 990-9585, Japan; 5Department of Neuro-Oncology/Neurosurgery, International Medical Center, Saitama Medical University, Hidaka 350-1298, Japan; 6Department of Neurosurgery, Faculty of Medicine, University of Tsukuba, Tsukuba 350-8576, Japan; 7Department of General Internal Medicine, National Cancer Center Hospital, Tokyo 104-0045, Japan; 8Department of Neurosurgery, Kyorin University Faculty of Medicine, Mitaka 181-8611, Japan

**Keywords:** metformin, newly diagnosed glioblastoma, dose-escalation, phase I, temozolomide

## Abstract

**Simple Summary:**

Glioblastoma is an incurable disease, demanding new therapeutic approaches. Our preclinical studies proved that the antidiabetic drug metformin could induce the differentiation of stem-like glioma-initiating cells and suppress tumor formation through AMPK-FOXO3 activation, suggesting the potential of metformin for treating glioblastoma. Taking into consideration that the anti-cancer effects of metformin are known to be dose-dependent, we conducted a dose-escalation phase I study to evaluate the safety and determine the recommended phase II dose of metformin in combination with maintenance temozolomide in patients with newly diagnosed glioblastoma. We show that the 2250 mg/day metformin appeared to be well tolerated with acceptable toxicity. Therefore, we proceed to a phase II study with 2250 mg/day metformin to evaluate its clinical benefits. Cancer stem/initiating cells are resistant to existing radiotherapy or chemotherapy; thus, our strategy targeting glioma-initiating cells using metformin is a novel therapeutic strategy which could possibly improve the outcome of glioblastoma.

**Abstract:**

Glioblastoma (GBM) inevitably recurs due to a resistance to current standard therapy. We showed that the antidiabetic drug metformin (MF) can induce the differentiation of stem-like glioma-initiating cells and suppress tumor formation through AMPK-FOXO3 activation. In this study, we design a phase I/II study to examine the clinical effect of MF. We aim to determine a recommended phase II MF dose with maintenance temozolomide (TMZ) in patients with newly diagnosed GBM who completed standard concomitant radiotherapy and TMZ. MF dose-escalation was planned using a 3 + 3 design. Dose-limiting toxicities (DLTs) were assessed during the first six weeks after MF initiation. Three patients were treated with 1500 mg/day MF and four patients were treated with 2250 mg/day MF between February 2021 and January 2022. No DLTs were observed. The most common adverse effects were appetite loss, nausea, and diarrhea, all of which were manageable. Two patients experienced tumor progression at 6.0 and 6.1 months, and one died 12.2 months after initial surgery. The other five patients remained stable at the last follow-up session. The MF dose of up to 2250 mg/day combined with maintenance TMZ appeared to be well tolerated, and we proceeded to a phase II study with 2250 mg/day MF.

## 1. Introduction

Glioblastoma (GBM) is the most common and aggressive primary brain tumor found in adult patients. The standard treatment for GBM includes maximal surgical resection followed by radiotherapy along with concomitant temozolomide (TMZ) chemotherapy and six maintenance cycles of TMZ therapy. This treatment regimen is applicable for patients aged ≤ 70 years with good general and neurological conditions [1]. Patient age, performance status, extent of resection, neurological function, isocitrate dehydrogenase (*IDH*) 1/2 mutations, and O^6^-methylguanine deoxyribonucleic acid methyltransferase (*MGMT*) promoter methylation status are the most consistently reported prognostic factors for patients with chemoradiotherapy-treated GBM [2,3,4].

The prognosis of patients with GBM is poor, with a 5-year survival rate of 15% in Japan [5]. Half of the patients die within 2 years because of inevitable and fatal tumor recurrence [3]. To achieve long-term survival and subsequently cure of the patients, the prevention of tumor recurrence is of utmost clinical importance; glioma stem/initiating cells are now considered to play a critical role in this control [6]. Cancer stem/initiating cells are a rare subpopulation of tumor cells capable of self-renewal and can generate differentiated progenies, which constitute the bulk of tumor mass [6,7]. In addition to the driving force for tumor growth and maintenance, cancer stem/initiating cells show increased resistance to existing anticancer therapies, including radiotherapy or chemotherapy [8,9]. Thus, developing a new targeted therapy for cancer stem/initiating cells could be a promising strategy to drastically improve the outcome of GBM.

Thus, we have been investigating the molecular mechanism involved in the maintenance and differentiation of cancer stem/initiating cells using glioma-initiating cells harboring stem-like properties directly derived from GBM tissues [6,10,11,12]. Phosphatidylinositol 3-kinase (PI3K) and mitogen-activated protein kinase (MAPK) pathways are involved in maintaining stem-like glioma-initiating cells; the inhibition of both pathways induces differentiation and suppresses their tumor-initiating potential more effectively than that achieved by the inhibition of a single pathway [10,11]. We also identified forkhead box O3 (FOXO3) as an integrator of the MAPK and PI3K pathways in the maintenance of stem-like glioma-initiating cells, and FOXO3 activation is sufficient to induce differentiation and suppress the tumor-initiating potential of stem-like glioma-initiating cells [12]. Thus, FOXO3 activators could be a potential molecular targeting drug for glioma-initiating cell-directed therapies. Drug screening identified metformin (MF), a biguanide and the most widely used drug for the treatment of type 2 diabetes, as an activator of FOXO3 in stem-like glioma-initiating cells [6]. MF activates FOXO3 via AMP-activated protein kinase (AMPK) and thereby induces differentiation of stem-like glioma-initiating cells in vitro, and effectively suppresses their tumor formation in vivo [6]. These findings from our preclinical studies provided a rationale to investigate the effects of MF on GBM in a clinical trial. Moreover, as previous studies have demonstrated that the anti-cancer effects of MF are significant at high doses [13,14,15,16,17], it is critical to determine the optimal MF dose to manifest anti-cancer activity in humans.

We therefore designed a phase I/II study to determine the maximum tolerated dose and examine the clinical effects of MF in combination with maintenance TMZ in non-diabetic patients with newly diagnosed GBM who have completed initial chemoradiation therapy with the smallest residual tumor burden possible. Here, we aim to present the result of this phase I dose-escalation trial to assess MF tolerability and feasibility and to establish the recommended phase II dose for the further evaluation of this therapeutic approach.

## 2. Materials and Methods

### 2.1. Study Design and Treatment

This was a multicenter single-arm phase I/II study conducted in Japan that was not randomized or blinded. The study protocol was approved by the National Cancer Center Hospital Certified Review Board (T2020002). All patients provided written informed consent. The study was conducted in accordance with the Declaration of Helsinki. The study is registered at https://jrct.niph.go.jp (accessed on 1 February 2021), under the identifier jRCTs031200326.

The study was composed of phase I and II trials, involving 22 patients. The phase I trial was a dose-escalation design with the aim to confirm the tolerability of MF combined with maintenance TMZ in patients with newly diagnosed GBM, and to determine the recommended dose of MF for phase II. The phase I trial was conducted at two sites (National Cancer Center Hospital and The University of Tokyo) between February 2021 and January 2022. The phase II trial is ongoing as a single arm design at five sites since April 2022, aimed to investigate efficacy with the primary endpoint of progression-free survival (PFS) rate 12 months after initial surgery. The current study report presents the results of the phase I trial.

All study participants received tumor resection or biopsy followed by standard concomitant chemotherapy with TMZ (75 mg/m^2^/day) and radiotherapy consisting of 60 Gy in 30 fractions. A dose escalation scheme with three different dose levels was designed; the target dose of MF was 1500 mg/day in the Level 1 cohort, 2250 mg/day in the Level 2 cohort, and 1000 mg/day in the Level −1 cohort (Table 1). MF treatment with Step 1 dose (described below in Table 1) was planned to be initiated within 3–8 weeks from the last day of concomitant TMZ and radiotherapy (Figure 1). Step 2 dose was initiated after one week of MF Step 1 dose treatment. After one week with the Step 2 dose, the treatment of MF with Step 3 dose for 28 days combined with standard maintenance TMZ (150 mg/m^2^/day for 5 days) (Cycle 1) was initiated. Then, 28 days after the start of Cycle 1, Cycle 2 maintenance therapy combining MF (Step 3 dose for 28 days) with TMZ (200 mg/m^2^/day for 5 days) was initiated, followed by Cycle 3–6, which repeated Cycle 2. After the completion of Cycle 6 of this maintenance therapy, MF alone (Step 3 dose) was continued for 365 days from the day of initiation (Figure 1).

In the phase I study, in the Level 1 cohort, MF was orally administered at a dose of 500, 1000, and 1500 mg/day in Step 1, 2, and 3, respectively. In the Level 2 cohort, it was administered at a dose of 500, 1500, and 2250 mg/day in Step 1, 2, and 3, respectively. Level −1 was defined in case Level 1 was not tolerated, and MF was administered at a dose of 500, 750, and 1000 mg/day in Step 1, 2, and 3, respectively (Table 1). Dose escalation from Level 1 to Level 2 proceeded according to a “3 + 3 design”, and intra-patient dose escalation was not allowed. The maximum dose in this study was set to 2250 mg/day MF, the maximum dose approved by Pharmaceuticals and Medical Devices Agency (PMDA) in Japan, based on the package insert. Treatment was continued until disease progression or clinically unacceptable toxicity was observed. In the phase II, MF would be administered at the dose determined in the phase I. If a patient experienced tumor progression, second-line chemotherapy was offered under the physician’s discretion. This study was determined to be terminated when one of the following adverse events was observed: grade ≥ 3 hypoglycemia, grade ≥ 3 lactic acidosis, or grade ≥ 3 rhabdomyolysis.

### 2.2. Patients

Male or female patients between 20 and 74 years of age with newly diagnosed supratentorial GBM histologically diagnosed according to the World Health Organization (WHO) 2016 classification [18] were considered eligible for the study. Regarding the genetic aberrations, we also reclassified these tumors into the latest classification according to the WHO 2021 classification [19]. Patients were required to have completed initial radiotherapy with concomitant TMZ therapy. The inclusion criteria were as follows: Karnofsky performance status ≥ 70, absolute neutrophil count ≥ 1500/mm^3^, hemoglobin ≥ 8.0 g/dL, platelets ≥ 100,000/mm^3^, aspartate aminotransferase (AST) ≤ 120 IU/L, or alanine aminotransferase (ALT) ≤ 120 IU/L. In addition, the estimated glomerular infiltration rate had to be ≥45 mL/min/1.73 m^2^ in the Level 1 cohort and ≥60 mL/min/1.73 m^2^ in the Level 2 cohort. Measurable or non-measurable disease was allowed. Exclusion criteria were as follows: patients with diabetes mellitus (defined as HbA1c level > 6.2% or routine use of insulin or glucose-lowering drug), history of biguanide drug allergy, lactic acidosis, unstable angina, pulmonary fibrosis, interstitial pneumonia, pulmonary emphysema, psychosis, cancer, or the routine use of immunosuppressive therapy for diseases other than brain tumor.

### 2.3. Genetic Analysis

In six patients who were treated in National Cancer Center Hospital, tumor DNA was extracted from frozen tumor tissues using a DNeasy Blood & Tissue Kit (Qiagen; Tokyo, Japan). The presence of hotspot mutations in the *IDH1* (R132) and *IDH2* (R172) genes was assessed by Sanger sequencing and/or pyrosequencing, as described previously [20,21]. Pyrosequencing assays were designed to detect all known mutations in these genes [20]. The two mutation hotspots in the *telomerase reverse transcriptase (TERT)* gene promoter were analyzed using Sanger sequencing and/or pyrosequencing, as reported previously [22]. The mutation hotspots at codons 27 and 34 of the histone H3.3 (*H3F3A*) gene, and those at codon 600 of the *B-Raf* (*BRAF*) gene were analyzed using Sanger sequencing and/or pyrosequencing [21]. The methylation status of the *O-6-methylguanine DNA methyltransferase* (*MGMT*) promoter was analyzed using the bisulfite modification of the tumor genomic DNA, followed by pyrosequencing, as previously described [22]. The *MGMT* promoter methylation status was defined as methylated when its mean level at the 16 CpG sites was 16% or greater than 16%, and unmethylated when less than 16% [23].

In one patient who was treated in The University of Tokyo (JP-02-001), the mutation status of *IDH1/2*, *TERT*, *H3F3A* and *BRAF* was obtained from comprehensive genomic profiling test using FoundationOne^®^ CDx Cancer Genomic Profile (Cambridge, MA, USA). *MGMT* promoter methylation status was assessed by methylation-specific polymerase chain reaction [24].

### 2.4. Study Endpoints

The primary endpoints were the evaluation of dose-limiting toxicity (DLT) in the phase I and progression-free survival rate at 12 months after initial surgery in the phase II. Adverse events (AEs) and biochemical (hematology, blood biochemistry, and urinalysis) abnormalities were assessed. AEs were assessed according to the National Cancer Institute–Common Terminology Criteria for Adverse Events v5.0 Japan Clinical Oncology Group version.

### 2.5. Dose-Limiting Toxicity

DLT evaluation period was set from the day of MF initiation to 28 days after the start of Cycle 1 of MF combined with maintenance TMZ. DLT was defined as grade 4 thrombocytopenia, grade 4 neutropenia with more than 7 days fever, grade 3 febrile neutropenia, grade ≥ 3 non-hematological toxicity other than hyponatremia, hypokalemia, manageable nausea, vomiting, diarrhea, grade ≥ 2 hypoglycemia, grade ≥ 3 lactic acidosis, or grade ≥ 2 rhabdomyolysis.

### 2.6. Statistical Analyses

PFS was calculated as the interval between the date of initial surgery and that of progression, death, or censorship at the last follow-up. OS was defined as the interval between the date of initial surgery and death or censorship at the last follow-up. Data management and data analysis were performed by WDB Clinical Research Co., Ltd, Tokyo, Japan.

## 3. Results

### 3.1. Patient Characteristics

Between February 2021 and January 2022, seven patients were enrolled in this study based on the inclusion and exclusion criteria. The first three patients were assigned to receive 1500 mg/day MF in the Level 1 cohort, and the remaining four patients received 2250 mg/day MF in the Level 2 cohort.

The median age of the patients was 41 years (range: 32–67 years). Three tumors (42.9%) were *IDH1/2* mutants and the remaining four (57.1%) were *IDH1/2* wild-types. All of the three tumors with *IDH1/2* mutations had necrosis and microvascular proliferation, thus our cohort included four patients with GBM, *IDH*-wildtype, WHO grade 4 (57.1%) and three patients with astrocytoma, *IDH*-mutant, WHO grade 4 (42.9%) based on the WHO 2021 classification. Four patients (57.1%) had no measurable contrast-enhancement lesion in post-operative magnetic resonance imaging. The *MGMT* promoter was methylated in the tumors of two patients (28.6%) and unmethylated in five patients (71.4%) (Table 2).

### 3.2. Safety Data

AEs are listed in Table 3. No DLTs were observed in either dose cohort during the DLT assessment period. However, we observed grade 1 diarrhea (n = 2, 28.6%) in the Level 1 cohort, and grade 2 leukopenia (n = 1, 14.3%), neutropenia (n = 1, 14.3%), lymphocytopenia (n = 1, 14.3%), grade 1 appetite loss (n = 2, 28.6%), nausea (n = 2, 28.6%), vomiting (n = 1, 14.3%), and constipation (n = 1, 14.3%) in the Level 2 cohort.

The most common AEs were appetite loss, nausea, and diarrhea, which were observed in three patients (42.9%). Other gastrointestinal symptoms, such as constipation, vomiting, and abdominal pain, were also observed. Hematological AEs included leukopenia, neutropenia, increased ALT in two patients (28.6%), and lymphocytopenia, thrombocytopenia, and increased AST in one patient (14.3%). The maximum concentrations of AST and ALT were 74 IU/L and 128 IU/L, respectively.

One patient (JP-02-001) in the Level 2 cohort manifested grade 2 diarrhea, appetite loss, and abdominal pain during treatment with MF Step 2 dose of 1500 mg/day before Cycle 1 of TMZ. After suspending MF treatment for one week, he met the start criteria of MF and TMZ; he was then administered MF Step 2 dose of 1500 mg/day combined with standard maintenance TMZ (150 mg/m^2^/day for 5 days). However, he continued to have grade 2 appetite loss for 2 weeks, for which a dose reduction to 1000 mg/day MF was required thereafter. This patient was then admitted due to grade 3 generalized seizure with small intratumoral hemorrhage, which was likely related to the tumor. He continued two more cycles of maintenance TMZ combined with MF at the dose of 1000 mg/day before he experienced disease progression. Our Data and Safety Committee excluded this patient from the DLT assessment, since he did not receive MF Step 3 dose of 2250 mg/day combined with standard maintenance TMZ.

There were no serious complications, such as hypoglycemia, lactic acidosis, and rhabdomyolysis.

### 3.3. Efficacy

Two patients experienced tumor progression 6.0 and 6.1 months after initial surgery. One patient died 12.2 months after initial surgery. The other five patients remained stable at the time of last follow-up (Figure 2). The PFS rate at 6 months was 85.7% at the end of June 2022. One patient with *IDH1/2* wild-type (JP-01-002) and one patient with *IDH1/2* mutant GBM (JP-01-001) completed the MF treatment for 1 year and are undergoing observation.

## 4. Discussion

This phase I study evaluated the tolerability and feasibility of MF in combination with maintenance TMZ in patients with newly diagnosed GBM. We found that the combination seemed to be well tolerated in this population and there was no DLT up to 2250 mg/day of MF, which is the maximum daily dose approved by the PMDA in Japan. Thus, the recommended dose of MF for the subsequent phase II was determined to be 2250 mg/day.

### 4.1. Safety

Regarding the safety profile, gastrointestinal symptoms, including diarrhea (42.9%), nausea (42.9%), appetite loss (42.9%), constipation (28.6%), and abdominal pain (14.3%), were commonly observed. All of them were manageable with grade ≤ 2. Diarrhea and abdominal pain were most likely related to MF [25,26], and constipation was most likely related to TMZ [27]. Nausea and appetite loss could be related to both MF and TMZ [25,26,27]. To minimize gastrointestinal symptoms, our protocol was designed to start with 500 mg/day MF for 1 week, which would then be titrated up if tolerated. Most patients received anti-emetic drugs (both granisetron and aprepitant) and anti-constipation drug (magnesium oxide). Four patients (57.1%) received anti-diarrheal drug (loperamide) provisionally. These supportive medications helped to relieve these symptoms, which were frequently observed in the first 42 days of the DLT evaluation period, and most of the patients improved after that period.

### 4.2. MF Dose

This study confirmed that 2250 mg/day MF was feasible and recommended this as the phase II dose. To the best of our knowledge, there are seven published phase I studies [26,28,29,30,31,32,33], which aimed to determine the maximum MF dose with various chemotherapy combinations. Fenn et al. evaluated eight patients with breast cancer receiving MF at a maximum daily dose of 2550 mg/day (850 mg three times a day) along with erlotinib at 150 mg daily and reported that two patients (25%) experienced grade 3 diarrhea. Thus, they concluded that the combination of erlotinib with 2550 mg/day MF was tolerated fairly well. They also emphasized the importance of titrating up administration of MF from a lower dose and early and aggressive supportive care [26]. Gulati et al. conducted a dose-determination clinical trial of MF along with chemoradiotherapy in 18 patients with head and neck squamous cell carcinoma and reported that 2550 mg/day MF is the highest dose with the tolerable profile [29]. In contrast, Proper et al. conducted a phase I study to determine the recommended phase II MF dose in patients with high-grade glioma. They evaluated six patients with newly diagnosed and seven with recurrent high-grade gliomas, and concluded that 1700 mg/day MF (850 mg twice a day) was acceptable, but 2550 mg/day MF (850 mg three times a day) was poorly tolerated due to nausea [33]. The absence of a dose-escalation scheme of MF or combination of MF with concomitant radiation therapy might have led to lower tolerability than that observed in this study. Maraka et al. conducted a phase I study of TMZ plus memantine, mefloquine, and MF as postradiation adjuvant therapy for newly diagnosed GBM, and reported that the dose of MF combined with TMZ had to be reduced from the original target dose of 2000 mg/day to 1700 mg/day because of nausea and dysgeusia [31]. In summary, 1700–2550 mg/day MF might be the maximum well-tolerated dose; thus, our recommended phase II dose of 2250 mg/day is in line with these previous reports.

### 4.3. MF Suppresses Tumorigenicity at a High Dose

MF dose setting is important because several studies have consistently suggested that the anti-cancer effects of MF are dose-dependent [13,14,15,16,17]. A subgroup analysis of a clinical study investigating MF in combination with erlotinib for treating advanced pancreatic cancer showed that patients with high plasma MF concentrations (>1 mg/L) seemed to have improved survival than those with lower concentrations, indicating the importance of administering a high MF dose [15]. Our previous experimental study showed that administering 500 mg/kg/day MF could improve survival time in intracranial stem-like glioma-initiating cell-implanted mice [6]. The MF dose of 500 mg/kg/day in a mouse can be translated to 45 mg/kg/day in a human [34]. If a patient’s body weight is 50 kg, 2250 mg/day MF corresponds to 45 mg/kg/day, which is equivalent to the dose used in our experiment. Thus, the MF dose of at least 2250 mg/day could be an optimal dose setting.

### 4.4. Anti-Cancer Mechanism of MF for Glioma Stem Cells

MF has been shown to possess antitumor activity [35,36,37,38,39]. Population-based studies have suggested that MF is associated with a reduced risk of cancer in patients with diabetes [35,36,38]. With respect to glioma, a population-based study suggested that the use of MF was associated with better overall and progression-free survival in patients with high-grade glioma [40], whereas a pooled analysis from several clinical trials in patients with GBM did not show a significant association between the use of MF and prolonged overall and progression-free survival [41]. The anti-cancer activity of MF includes the inhibition of complex I of the mitochondria oxidative phosphorylation chain, activation of AMPK, suppression of the mTORC1 pathways, and stimulation of the immune system [37]. In pre-clinical studies on glioblastoma, the potential effect of MF has been described in terms of the inhibition of tumor cell proliferation, differentiation and invasiveness, increase in apoptosis, and synergistic increase in sensitivity to chemoradiation therapy with TMZ [39]. Moreover, MF has been shown to inhibit endothelial cell proliferation at least partially by activating AMPK, suggesting its therapeutic potential as an anti-angiogenic agent [42].

Since cancer stem/initiating cells are presumably resistant to existing anticancer therapies, our strategy of using MF to target such a therapy-resistant population of cancer cells could possibly be a breakthrough in the treatment of intractable cancers, such as GBM. We assumed the following two mechanisms of MF on cancer stem/initiating cells: differentiation-inducing and anti-proliferative mechanisms. The induction of differentiation is caused by MF activating FOXO3 through AMPK activation, leading to the loss of stem cell capacity [6]. This was also supported by a previous study, which showed that MF use combined with TMZ significantly reduced sox 2 levels [43]. The inhibition of proliferation is caused by MF exerting cytostatic effects at low doses and cytotoxic effects at high doses on cancer stem cells through the decreased Akt activation [44]. Both these mechanisms contribute to the depletion of cancer stem cell population, minimizing the risk of tumor recurrence from cancer stem/initiating cells. The induction of differentiation could be obtained using 1 mM MF, whereas the inhibition of proliferation is observed using 10 mM MF [6,44]. Based on these considerations, the induction of differentiation might be the primary effects on cancer stem/initiating cells in the human body. Moreover, MF can easily cross the blood-brain barrier and accumulate in the brain parenchyma [6,45]. MF has been widely and safely used as an economical anti-diabetic drug [37], which is an advantage of repurposing MF as a drug for GBM treatment.

### 4.5. The Effect of MF Depends on Extracellular Glucose Concentration

Extracellular glucose concentration is critical for MF to exhibit its anti-cancer activity [6]. Our previous studies have shown that in vitro sphere formation is significantly inhibited by MF in low glucose concentrations of the culture medium but not in high glucose concentrations [6]. Additionally, the survival time of the tumor-bearing mice was significantly longer on implantation with stem-like glioma-initiating cells treated with MF in a low glucose concentration than on implantation with those treated with MF in a high glucose concentration [6]. The glucose concentration of the culture medium used in stem cell culture experiment is much higher than that of human body fluids [6,46]; therefore, we expect that MF could have potent inhibitory effects on cancer stem/initiating cells when MF is clinically administered to patients in normal glycemic condition. Possibly in line with this idea, the use of metformin in supposedly diabetic GBM patients was not significantly associated with survival in the pooled analysis of clinical trials mentioned above [41].

### 4.6. MF Has Limited Suppressive Effects on Massive Tumors or Progressively Growing Tumors

There have been several phase II studies to evaluate the anti-cancer activity of MF in various cancers with varied findings [15,25,47,48,49,50,51,52,53,54]. Brown et al. reported that MF is associated with better-than-expected overall survival in 38 patients with epithelial ovarian cancer. They administrated MF (1000 or 2000 mg/day) before primary debulking surgery, followed by adjuvant chemotherapy with MF (1000 or 2000 mg/day). Their translational studies showed a significant reduction in the number of cancer stem cells with MF treatment, suggesting MF’s effect on cancer stem cells [25]. Marrone et al. showed a significant benefit in PFS with the use of MF in 25 patients with non-small cell lung carcinoma (NSCLC). They used 2000 mg/day MF combined with paclitaxel/carboplatin/bevacizumab [47]. In contrast, several randomized phase II clinical trials have not exhibited therapeutic effects with 2000 mg/day MF with chemoradiation in patients with NSCLC or pancreatic cancer [15,52,53].

These contrasting results could be attributed to the anti-cancer mechanism of MF that acts specifically on cancer stem cells, but has limited suppressive effects on massive or progressively growing tumors, which consist largely of non-stem tumor cells [6,25]. Therefore, the therapeutic effect of MF, which is expected to prevent tumor recurrence from cancer stem cells, could be best explored when it is used in minimal residual tumors. In this hypothesis, our phase II study is aimed at patients with newly diagnosed GBM who have undergone initial resection and completed chemoradiation therapy, and uses PFS at 12 months as a primary endpoint to evaluate the therapeutic effect of MF. Although we observed that five patients remained stable, with a PFS rate at 6 months of 85.7% at the time of data analysis, further study is needed to evaluate the efficacy of MF in the phase II study.

There are some limitations to this phase I study. First, the small sample size in this study limited the ability to draw a firm conclusion regarding the safety of MF in combination with TMZ. We acknowledge the importance of careful monitoring in the phase II study.

## 5. Conclusions

In conclusion, this phase I study demonstrated that 2250 mg/day MF combined with TMZ appeared to be well tolerated in patients with newly diagnosed GBM. The commonly observed AEs were gastrointestinal symptoms, including diarrhea, nausea, and appetite loss. However, all of these were grade ≤ 2 and could be accepted. Based on the findings of this phase I study, we proceeded to the phase II study with a MF dose of 2250 mg/day.

## 6. Patents

The Japanese Unexamined Patent Publication number is P2014-221752A.

## Figures and Tables

**Figure 1 cancers-14-04222-f001:**
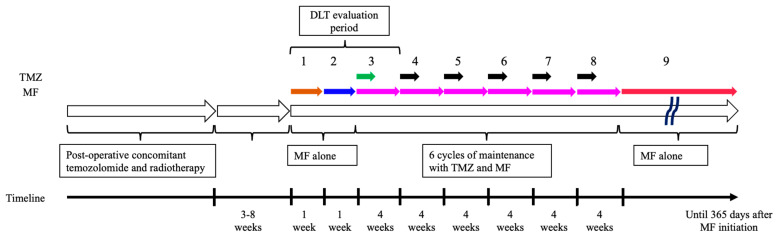
Treatment schedule. Treatment with metformin (MF) Step 1 dose was to be initiated within 3 to 8 weeks from the last day of concomitant temozolomide (TMZ) and radiotherapy. Dose-limiting toxicity (DLT) was evaluated from the day of MF initiation to 28 days after the start of Cycle 1 involving (MF) and (TMZ). The numbers 1–9 indicate the following: 1. MF alone with Step 1 dose for 7 days (Brown arrow); 2. MF alone with Step 2 dose for 7 days (Blue arrow); 3. Maintenance therapy Cycle 1: MF with Step 3 dose for 28 days (Pink arrow) and TMZ (150 mg/m^2^/day) for 5 days (Green arrow); 4. Maintenance therapy Cycle 2: MF with Step 3 dose for 28 days (Pink arrow) and TMZ (200 mg/m^2^/day) for 5 days (Black arrow); 5. Maintenance therapy Cycle 3: MF with Step 3 dose for 28 days (Pink arrow) and TMZ (200 mg/m^2^/day) for 5 days (Black arrow); 6. Maintenance therapy Cycle 4: MF with Step 3 dose for 28 days (Pink arrow) and TMZ (200 mg/m^2^/day) for 5 days (Black arrow); 7. Maintenance therapy Cycle 5: MF with Step 3 dose for 28 days (Pink arrow) and TMZ (200 mg/m^2^/day) for 5 days (Black arrow); 8. Maintenance therapy Cycle 6: MF with Step 3 dose for 28 days (Pink arrow) and TMZ (200 mg/m^2^/day) for 5 days (Black arrow); 9. MF alone with Step 3 dose until 365 days from MF initiation (Red arrow).

**Figure 2 cancers-14-04222-f002:**
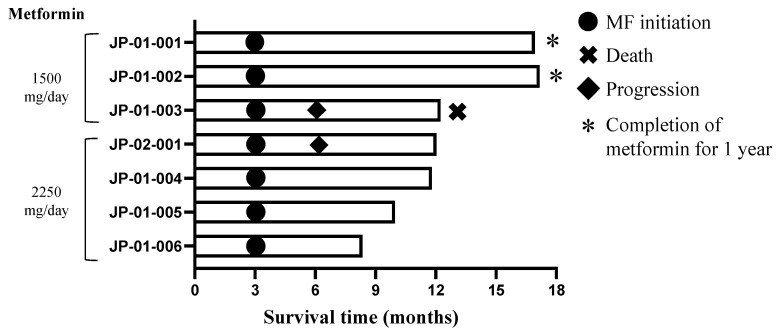
Swimmer plot of survival time from initial surgery. JP-01-001, 002, and 003 belonged to the Level 1 cohort (1500 mg/day MF), and the other JP-01-004, 005, and 006 and JP-02-001 to the Level 2 cohort (2250 mg/day MF), Two patients (JP-01-003 and JP-02-001) experienced tumor progression 6.1 months and 6.0 months after initial surgery, respectively. One patient (JP-01-003) died 12.2 months after initial surgery. The remaining five patients maintained stable disease at the time of last follow-up. Two patients completed MF therapy for 1 year (*). MF: metformin.

**Table 1 cancers-14-04222-t001:** MF dose-escalation scheme.

Dose Level	MF Dose		
	Step 1	Step 2	Step 3
Level 2	500 mg/day	1500 mg/day	2250 mg/day
Level 1	500 mg/day	1000 mg day	1500 mg/day
Level −1	500 mg/day	750 mg/day	1000 mg/day

MF: metformin.

**Table 2 cancers-14-04222-t002:** Characteristics of the 7 patients.

Dose Level	Patient	Sex	Age	KPS	Diagnosis (WHO2016)	Diagnosis (WHO2021)	Measurable Residual Tumor	*IDH1/2*	*MGMT*	*TERT*	*H3F3A*	*BRAF*
Level 1	JP-01-001	Male	32	90	Glioblastoma, *IDH*-mutant	Astrocytoma, *IDH*-mutant, WHO grade 4	No	Mutant	Unmethylated	Wild type	Wild type	Wild type
Level 1	JP-01-002	Female	67	80	Glioblastoma, *IDH*-wildtype	Glioblastoma, *IDH*-wildtype, WHO grade 4	No	Wild-type	Unmethylated	Mutant	Wild type	Wild type
Level 1	JP-01-003	Male	34	70	Glioblastoma, *IDH*-wildtype	Glioblastoma, *IDH*-wildtype, WHO grade 4	No	Wild-type	Unmethylated	Wild type	Wild type	Wild type
Level 2	JP-02-001	Male	46	100	Glioblastoma, *IDH*-wildtype	Glioblastoma, *IDH*-wildtype, WHO grade 4	Yes	Wild-type	Unmethylated	Wild type	Wild type	Wild type
Level 2	JP-01-004	Male	40	90	Glioblastoma, *IDH*-mutant	Astrocytoma, *IDH*-mutant, WHO grade 4	Yes	Mutant	Methylated	Wild type	Wild type	Wild type
Level 2	JP-01-005	Female	41	80	Glioblastoma, *IDH*-mutant	Astrocytoma, *IDH*-mutant, WHO grade 4	Yes	Mutant	Methylated	Wild type	Wild type	Wild type
Level 2	JP-01-006	Male	46	90	Glioblastoma, *IDH*-wildtype	Glioblastoma, *IDH*-wildtype, WHO grade 4	No	Wild-type	Unmethylated	Mutant	Wild type	Wild type

KPS Karnofsky performance status, *IDH1/2* Isocitrate dehydrogenase, *MGMT* O6-methyguanine deoxyribonucleic acid methyltransferase, *TERT* telomerase reverse transcriptase, *H3F3A* histone H3.3, *BRAF* B-Raf, PFS Progression free survival, OS Overall survival.

**Table 3 cancers-14-04222-t003:** Adverse events of any grade.

Category	All (n = 7)		Level 1: MF 1500 mg/Day (n = 3)			Level 2: MF 2250 mg/Day (n = 4)		
	All Grades	Grade 3	Grade 1	Grade 2	Grade 3	Grade 1	Grade 2	Grade 3
Hematologic								
Leukopenia	2 (28.6%)	0	1 (33.3%)	0	0	0	1 (25.0%)	0
Neutropenia	2 (28.6%)	0	0	1 (33.3%)	0	0	1 (25.0%)	0
Lymphocytopenia	1 (14.3%)	0	0	0	0	0	1 (25.0%)	0
Thrombocytopenia	1 (14.3%)	0	0	0	0	1 (25.0%)	0	0
Anemia	0	0	0	0	0	0	0	0
Hepatic								
Aspartate transaminase	1 (14.3%)	0	1 (33.3%)	0	0	0	0	0
Alanine transaminase	2 (28.6%)	0	0	1 (33.3%)	0	1 (25.0%)	0	0
Appetite loss	3 (42.9%)	0	0	0	0	2 (50.0%)	1 (25.0%)	0
Nausea	3 (42.9%)	0	0	0	0	3 (75.0%)	0	0
Vomiting	1 (14.3%)	0	0	0	0	1 (25.0%)	0	0
Constipation	2 (28.6%)	0	1 (33.3%)	0	0	1 (25.0%)	0	0
Fatigue	1 (14.3%)	0	1 (33.3%)	0	0	0	0	0
Diarrhea	3 (42.9%)	0	2 (66.7%)	0	0	0	1 (25.0%)	0
Abdominal pain	1 (14.3%)	0	0	0	0	0	1 (25.0%)	0
Seizure	1 (14.3%)	1 (14.3%)	0	0	0	0	0	1 (25.0%)
Somnolence	1 (14.3%)	0	0	0	0	1 (25.0%)	0	0
Intracranial hemorrhage	1 (14.3%)	0	0	0	0	1 (25.0%)	0	0

MF: metformin.

## Data Availability

The data presented in this study are available upon request from the corresponding author.

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
