# Peer review of "Metformin with Temozolomide for Newly Diagnosed Glioblastoma: Results of Phase I Study and a Brief Review of Relevant Studies"

_cancers, 2022, doi:10.3390/cancers14174222_

Round 1

Reviewer 1 Report

The authors present an interesting scientific paper regarding a Phase I study to examine the tolerability of Metformin treatment in newly diagnosed Glioblastoma patients.

Metformin (N, N-dimethylbiguanide) is the most widely used antihyperglycemic drug worldwide, being the current first-line therapy for all newly diagnosed type 2 diabetes (T2DM) patients. Recent epidemiological studies have confirmed that the administration of metformin to diabetic patients at the standard clinical dose (1,500-2,250 mg/day) is able to reduce the incidence of cancer and / or the relative mortality, even in patients with gliomas [Mazurek et al., 2020]. The experimental data also confirmed the activity of metformin in halting the progression of cancer, including cancer of the pancreas, prostate, stomach, breast and uterus, both alone and in combination with radiotherapy, and also in GBM [Hirsch, H.A.; Iliopoulos, D.; Tsichlis, P.N.; Struhl, K. Metformin selectively targets cancer stem cells, and acts together with chemotherapy to block tumor growth and prolong remission. Cancer Res. 2009, 69, 7507–7511, doi:10.1158/0008-5472.CAN-09-2994; Guarnaccia L, Marfia G, Masseroli MM, Navone SE, Balsamo M, Caroli M, Valtorta S, Moresco RM, Campanella R, Garzia E, Riboni L, Locatelli M. Frontiers in Anti-Cancer Drug Discovery: Challenges and Perspectives of Metformin as Anti-Angiogenic Add-On Therapy in Glioblastoma. Cancers (Basel). 2021 Dec 27;14(1):112. doi: 10.3390/cancers14010112. PMID: 35008275; PMCID: PMC8749852.)

This work represents an interesting contribution to the field of novel therapeutic strategies for GBM.

The design of the study is appropriate and the results are clearly reported. In addition, the references reported are appropriate.

Major comments:

-        It would be of interest to describe the molecular profile of each patient

-        In addition of the several studies related to the effect of MF on cancer stem cells, in particular on glioma-initiating cells, It has also been reported the anti-tumoral effect of MF on tumor endothelial cells. It would be of interest to discuss this aspect in discussion section.

Minor comments:

The authors should correct the posology unit of measurement in section 4.6 (lines 346 and 348).

Author Response

We sincerely appreciate all the three reviewers for taking the time to review our manuscript and giving us valuable comments.

We revised the manuscript according to the reviewers’ comments and showed them point by point.

We believe our manuscript is better than the previous one.

It would be of interest to describe the molecular profile of each patient

Response: We described molecular information about TERT, BRAF, H3F3A in the Table 2, which was changed to patient-specific manner.

In addition of the several studies related to the effect of MF on cancer stem cells, in particular on glioma-initiating cells, it has also been reported the anti-tumoral effect of MF on tumor endothelial cells. It would be of interest to discuss this aspect in discussion section. 

Response: We appreciate this reviewer’s comment and added “Moreover, MF has shown to inhibit endothelial cell proliferation at least partially by activating AMPK, suggesting its therapeutic potential as an anti-angiogenic agent” in lines 345-347.

Minor comments:

The authors should correct the posology unit of measurement in section 4.6 (lines 346 and 348).

Response: We apologize for these mistakes and corrected in the revised manuscript.

Reviewer 2 Report

This paper describes a phase I study to evaluate the dose of metformin with temozolomide in patients with newly diagnosed glioblastoma. In this study, the authors performed dose escalation to determine the maximum tolerated dose for metformin. Although the number of patients included in the study is relatively small, the analysis of the use of metformin in combination with TMZ treatment is a novelty. Nonetheless, some minor revisions are needed in the presentation of methods and results (see comments below).

Please merge the paragraphs in lines 56-60 and lines 61-67 for better reading flow.

Please include further references for the role of metformin in the therapy of glioblastoma (for example https://doi.org/10.3390/cancers14010112)

Line 112 ff.: Please explain the different cohorts (Level2, Level1 and Level -1) within the paragraph describing the content of table 1.

Figure 1: Please revise figure 1 for better overall understanding. It would be good to include a timeline (days or weeks) and a color code for the different doses of metformin

Line 182: Do not use abbreviations in headings. Please replace “DLT” with dose limiting toxicity

Table 2: Please revise table 2. Despite the given averages of e.g. IDH1/2 mutation status for the patient cohort, it would be good to include patient-specific information. Furthermore, the description "No of Patients” is not fitting the value age and the explanation of the abbreviation MF is missing in the footnote.

Line 261-263: “Diarrhea and abdominal pain might most likely be related to MF, and constipation might most likely be related to TMZ. Nausea and appetite loss could be related to both MF and TMZ.” A reference of this assumption is required here.

Please correct “al” into “al.” in line 283

Line 310: Please delete the sentence “Our study is unique in targeting cancer stem/initiating cells” There are several studies investigating the influence of MF on cancer stem cells. No experiments were conducted according to possible metformin-induced differences in the amount of cancer stem cells in the glioblastomas were made in this study.

Line 313: Please correct “differentiation inducing” to “induction of differentiation” or change the sentence.

Lines 304 – 325 (Section 4.4.): This section in general is worded too strongly as the authors did not checked experimentally for the influence of metformin on the amount of cancer stem cells in glioblastoma. Here, only assumptions can be drawn based on preclinical analysis. Please rephrase this paragraph accordingly.

Line 334: “The glucose concentration of the culture medium used in stem cell 333 culture experiment is much higher than that of human body fluids” A reference is required here.

Author Response

We sincerely appreciate all the three reviewers for taking the time to review our manuscript and giving us valuable comments.

We revised the manuscript according to the reviewers’ comments and showed them point by point.

We believe our manuscript is better than the previous one.

Please merge the paragraphs in lines 56-60 and lines 61-67 for better reading flow.

Response: We appreciate this reviewer’s comment and corrected according to this comment.

Please include further references for the role of metformin in the therapy of glioblastoma (for example https://doi.org/10.3390/cancers14010112)

Response: We appreciate this reviewer’s comment and added “In pre-clinical studies on glioblastoma, potential effect of MF has been described in terms of inhibition of tumor cell proliferation, differentiation and invasiveness, increase in apoptosis, and synergistic increase in sensitivity to chemoradiation therapy with TMZ” in lines 342-345.

Line 112 ff.: Please explain the different cohorts (Level2, Level1 and Level -1) within the paragraph describing the content of table 1.

Response: According to this comment, we added “Dose escalation scheme with three different dose levels was designed; the target dose of MF was 1,500mg/day in the Level 1 cohort, 2,250mg/day in the Level 2 cohort, and 1,000 mg/day in the Level -1 cohort (Table 1)” in lines 113-115.

Figure 1: Please revise figure 1 for better overall understanding. It would be good to include a timeline (days or weeks) and a color code for the different doses of metformin

Response: We appreciate this reviewer’s comment. We added timeline and changed color code for the different dose of metformin to improve readers’ understanding.

Line 182: Do not use abbreviations in headings. Please replace “DLT” with dose limiting toxicity

Response: We appreciate this reviewer’s comment and corrected according to this comment.

Table 2: Please revise table 2. Despite the given averages of e.g. IDH1/2 mutation status for the patient cohort, it would be good to include patient-specific information. Furthermore, the description "No of Patients” is not fitting the value age and the explanation of the abbreviation MF is missing in the footnote.

Response: We appreciate this reviewer’s comment. We revised Table 2 to include patient-specific information.

Line 261-263: “Diarrhea and abdominal pain might most likely be related to MF, and constipation might most likely be related to TMZ. Nausea and appetite loss could be related to both MF and TMZ.” A reference of this assumption is required here.

Response: We appreciate this reviewer’s comment. We added references to each of the assumption.

Please correct “al” into “al.” in line 283

Response: We corrected this according to this comment.

Line 310: Please delete the sentence “Our study is unique in targeting cancer stem/initiating cells” There are several studies investigating the influence of MF on cancer stem cells. No experiments were conducted according to possible metformin-induced differences in the amount of cancer stem cells in the glioblastomas were made in this study.

Response: We agree with this reviewer’s comment and deleted the sentence “Our study is unique in targeting cancer stem/initiating cells” in the revised manuscript.

Line 313: Please correct “differentiation inducing” to “induction of differentiation” or change the sentence.

Response: We corrected this according to this comment.

Lines 304 – 325 (Section 4.4.): This section in general is worded too strongly as the authors did not checked experimentally for the influence of metformin on the amount of cancer stem cells in glioblastoma. Here, only assumptions can be drawn based on preclinical analysis. Please rephrase this paragraph accordingly.

Response: We appreciate this reviewer’s comment. We rephrased this paragraph to describe mildly. For example, the phrase “our strategy could be a breakthrough” was changed into “our strategy of using MF to target such a therapy-resistant population of cancer cells could possibly be a breakthrough”.

Line 334: “The glucose concentration of the culture medium used in stem cell culture experiment is much higher than that of human body fluids” A reference is required here. 

Response: We added references according to this comment.

Reviewer 3 Report

Ohno and colleagues describe the results of phase I clinical trial employing metformin and temozolomide (MF plus TMZ) application to de novo glioma patients, post completion of the chemotherapy phase. The main goal of the phase I trial was to evaluate maximal dose of metformin which can be safely applied as an adjuvant to maintenance temozolomide. Authors describe the scheme of metformin dose titration and list dose-limiting toxicities of two drugs combination. They define safe maximal dose of metformin which can be applied together with temozolomide as 2250 mg per day. Based on this observation they proceed to phase II study, aiming to evaluate MF plus TMZ regimen therapeutic efficacy. Phase II is ongoing, and its results are yet to be known.

Additionally, the rationale for using metformin as anti-glioblastoma drug is given, based on previous work from the same group demonstrating that metformin targets glioblastoma stem cells and induces their differentiation, which translates to reduced aggressiveness of the tumor when implanted into the experimental animal.

The obtained results are interesting and very promising. There are only some minor issues to be corrected, all in regard to the data description and the relevant literature overview.

1.       The patients selected for the trial have been diagnosed with IDH1/2 wt (4 patients) and IDH1/2 mutant (3 patients) type of glioma. According to the fifth edition of WHO classification of CNS tumors (2021), glioblastoma definition is now limited to IV grade tumor without IDH1/2 mutation, and IV grade tumors with IDH1/2 mutation are currently defined as IV grade astrocytoma with IDH1/2 mutations (https://pubmed.ncbi.nlm.nih.gov/34185076/). The most recent classification needs to be applied in the manuscript, and in the future. IDH1/2 status is known to affect chemotherapy outcome, and the possibility that proposed MF plus TMZ regimen efficacy is affected by IDH1/2 mutation needs to be taken into consideration.

2.       MF plus TMZ was previously applied to glioma patients. Study has been mentioned in the References (Maraka et al., 2019), however hasn’t been sufficiently discussed.

3.       Line 239 versus line 538: there are to different values for 6 month PFS – please explain the difference or correct if applicable.

4.       Figure 2: How exactly the survival time was measured? From the moment of the initial surgery or the start of MF application? If the time “0” indicates surgery, could Authors mark on the graph when MF treatment was initiated in each patient?

5.       Line 240. One patient with IDH1/2 wild-type and one patient with IDH1/2 mutant GBM have completed the MF treatment for 1 year and have followed-up without any additional treatment drug. Does “without any additional treatment drug” simply mean that patients do not receive any treatment, just undergo the observation?

6.       Differentiation of glioma stem like cells makes them more sensitive to temozolomide. It has been shown previously in vitro and in vivo (see Valtarte et al.  https://pubmed.ncbi.nlm.nih.gov/34012924/ etc). These literature data support the concomitant MF plus TMZ application to glioma patients, and are worth to be cited here.

7.       The reformulation of the paper title is strongly advised. First, the clou of the current phase I study was metformin dose escalation, and thorough DLTs characterization. Second, mentioning cancer stem cells in the title is not necessary and actually brings more confusion than benefit. The rationale of applying metformin is well explained in the abstract and manuscript body.

Author Response

We sincerely appreciate all the three reviewers for taking the time to review our manuscript and giving us valuable comments.

We revised the manuscript according to the reviewers’ comments and showed them point by point.

We believe our manuscript is better than the previous one.

The patients selected for the trial have been diagnosed with IDH1/2 wt (4 patients) and IDH1/2 mutant (3 patients) type of glioma. According to the fifth edition of WHO classification of CNS tumors (2021), glioblastoma definition is now limited to IV grade tumor without IDH1/2 mutation, and IV grade tumors with IDH1/2 mutation are currently defined as IV grade astrocytoma with IDH1/2 mutations (https://pubmed.ncbi.nlm.nih.gov/34185076/). The most recent classification needs to be applied in the manuscript, and in the future. IDH1/2 status is known to affect chemotherapy outcome, and the possibility that proposed MF plus TMZ regimen efficacy is affected by IDH1/2 mutation needs to be taken into consideration. 

Response: We appreciate this reviewer’s comment, and described the diagnosis based on the WHO 2021 classification in addition to the WHO 2016 classification in the revised manuscript.

MF plus TMZ was previously applied to glioma patients. Study has been mentioned in the References (Maraka et al., 2019), however hasn’t been sufficiently discussed.

Response: We appreciate this reviewer’s comment and added “Maraka et al. conducted a phase I study of TMZ plus memantine, mefloquine, and MF as postradiation adjuvant therapy for newly diagnosed GBM, and reported that the dose of MF combined with TMZ had to be reduced from the original target dose of 2,000mg/day to 1,700 mg/day because of nausea and dysgeusia” in lines 318-321.

Line 239 versus line 538: there are to different values for 6 month PFS – please explain the difference or correct if applicable. 

Response: We apologize for this mistake and corrected in the revised manuscript.

Figure 2: How exactly the survival time was measured? From the moment of the initial surgery or the start of MF application? If the time “0” indicates surgery, could Authors mark on the graph when MF treatment was initiated in each patient?

Response: We appreciate this reviewer’s comment. As the reviewer’s comment, “0” indicate the date of initial surgery. We marked the “MF initiation” in the Fig.2 in the revised manuscript. We also described the definition of OS in 2.6 Statistical analysis in the Method section in line 212-213.

Line 240. One patient with IDH1/2 wild-type and one patient with IDH1/2 mutant GBM have completed the MF treatment for 1 year and have followed-up without any additional treatment drug. Does “without any additional treatment drug” simply mean that patients do not receive any treatment, just undergo the observation?

Response: We appreciate this reviewer’s comment and changed as following “they are undergoing observation” in line 269-270.

Differentiation of glioma stem like cells makes them more sensitive to temozolomide. It has been shown previously in vitro and in vivo (see Valtarte et al.  https://pubmed.ncbi.nlm.nih.gov/34012924/ etc). These literature data support the concomitant MF plus TMZ application to glioma patients, and are worth to be cited here.

Response: We appreciate this reviewer’s comment and added “This was also supported by a previous study, which showed that MF use combined with TMZ significantly reduced sox 2 levels” in line 354-355.

The reformulation of the paper title is strongly advised. First, the clou of the current phase I study was metformin dose escalation, and thorough DLTs characterization. Second, mentioning cancer stem cells in the title is not necessary and actually brings more confusion than benefit. The rationale of applying metformin is well explained in the abstract and manuscript body.

Response: We appreciate this reviewer’s comment and changed the title into “Metformin with Temozolomide for Newly diagnosed Glioblastoma: Results of Phase I Study and a Brief Review of Relevant Studies” in the revised manuscript.